# Brigatinib for Pretreated, ALK-Positive, Advanced Non-Small-Cell Lung Cancers: Long-Term Follow-Up and Focus on Post-Brigatinib Lorlatinib Efficacy in the Multicenter, Real-World BrigALK2 Study

**DOI:** 10.3390/cancers14071751

**Published:** 2022-03-30

**Authors:** Renaud Descourt, Maurice Pérol, Gaëlle Rousseau-Bussac, David Planchard, Bertrand Mennecier, Marie Wislez, Jacques Cadranel, Alexis Benjamin Cortot, Florian Guisier, Loïck Galland, Pascal Do, Roland Schott, Éric Dansin, Jennifer Arrondeau, Jean-Bernard Auliac, Margaux Geier, Christos Chouaïd

**Affiliations:** 1Institut de Cancérologie, CHU Augustin-Morvan, 2 Avenue Foch, CEDEX, 29200 Brest, France; margaux.geier@chu-brest.fr; 2Department of Medical Oncology, Centre Léon-Bérard, 69373 Lyon, France; maurice.perol@lyon.unicancer.fr; 3Centre Hospitalier Intercommunal, 40 Avenue de Verdun, CEDEX, 94000 Créteil, France; gaelle.rousseaubussac@chicreteil.fr (G.R.-B.); j-b.auliac@chicreteil.fr (J.-B.A.); christos.chouaid@chicreteil.fr (C.C.); 4Hôpital Gustave-Roussy, 39 Rue Camille-Desmoulins, CEDEX, 94805 Villejuif, France; david.planchard@gustaveroussy.fr; 5CHU Strasbourg, 1 pl Hôpital, 67000 Strasbourg, France; bertrand.mennecier@chru-strasbourg.fr; 6Assistance Publique–Hôpitaux de Paris, Hôpital Cochin, 27 Rue du Fg Saint-Jacques, CEDEX 14, 75679 Paris, France; marie.wislez@aphp.fr (M.W.); jennifer.arrondeau@aphp.fr (J.A.); 7Assistance Publique–Hôpitaux de Paris, Hôpital Tenon, 4 Rue de la Chine, CEDEX 20, 75970 Paris, France; jacques.cadranel@aphp.fr; 8CHU Lille, Boulevard du Pr Jules Leclerc, 59000 Lille, France; alexis.cortot@chru-lille.fr; 9CHU Rouen, 37 Bd Gambetta, 76000 Rouen, France; florian.guisier@chu-rouen.fr; 10Centre Georges-François-Leclerc, 1 Rue du Professeur Marion, 21000 Dijon, France; lgalland@cgfl.fr; 11Centre François-Baclesse, 3 Avenue du Général Harris, 14000 Caen, France; pdo@baclesse.unicancer.fr; 12Institut de Cancérologie Strasbourg Europe, 17 Rue Albert Calmette, 67200 Strasbourg, France; r.schott@icans.eu.fr; 13Centre Oscar-Lambret, 3 Rue Frédéric Combemale, 59000 Lille, France; e-dansin@o-lambret.fr

**Keywords:** NSCLC, ALK rearrangment, brigatinib, lorlatinib

## Abstract

**Simple Summary:**

This analysis assesses the efficacy of brigatinib, a next-generation ALK inhibitor in ALK^+^ advanced non-small cell lung cancer (aNSCLC) included in the brigatinib French Early-Access Program (1 August 2016–21 January 2019), with a focus on post-brigatinib lorlatinib efficacy. With a median follow-up of 40.4 months (95% CI, 38.4–42.4), the median investigator-assessed PFS of the 183 included patients was 7.4 months (5.9–9.6) and overall survival from brigatinib initiation was 20.3 (15.6–27.6) months. For patients who received 1 (*n* = 23), 2 (*n* = 146) or 3 (*n* = 14) ALKi(s) before brigatinib, the median overall survival was 33 (9.7—not reached), 20.3 (15.7–28.7) and 18.1 (3.3–24.5) months, respectively. Ninety-two (50.3%) patients received one agent(s) post-brigatinib; 68 (73.9%) of them received lorlatinib: 51 (75%) immediately post-brigatinib. With a median follow-up of 29.9 months (25.7–33.1), the median overall survival from lorlatinib initiation was 14.1 months (10.3–19.2). Analysis results confirmed brigatinib effectiveness in a population of heavily pretreated ALK^+^ positive aNSCLC patients and the activity of lorlatinib after brigatinib. We confirm that neither the manuscript nor any parts of its content are currently under consideration or published in another journal. All authors have approved the manuscript and agree with its submission to cancers.

**Abstract:**

Brigatinib is a next-generation ALK inhibitor (ALKi) that shows efficacy in ALK inhibitor naïve and post-crizotinib ALK^+^ advanced NSCLCs (aNSCLCs). The efficacy of brigatinib was retrospectively assessed in patients with aNSCLCs included in the brigatinib French Early-Access Program (1 August 2016–21 January 2019). The primary endpoint was investigator-assessed progression-free survival (invPFS) and the primary analysis was updated in 2021 with a longer follow-up, focused on post-brigatinib lorlatinib efficacy. Sixty-six centers included 183 patients: median age 60 ± 12.7 years; 78.3% never/former smokers; median of 3 ± 1 previous lines and 2 ± 0.5 ALKis; 37.1% ECOG PS 2 and 55.6% >3 metastatic sites. The median follow-up from brigatinib initiation was 40.4 months (95% CI 38.4–42.4). InvPFS was 7.4 months (95% CI 5.9–9.6), median duration of treatment (mDOT) was 7.3 months (95% CI 5.8–9.4) and median overall survival (mOS) was 20.3 months (95% CI 15.6–27.6). The median DOT and OS from brigatinib initiation tend to decrease with the number of ALK inhibitors used in previous lines of therapy. Based on the data collected, 92 (50.3%) patients received ≥1 agent(s) post-brigatinib and 68 (73.9%) of them received lorlatinib, with 51 (75%) immediately receiving it post-brigatinib, 12 (17.6%) receiving it after one and 5 (7.4%) after ≥2 subsequent treatments. The median follow-up was 29.9 (95% CI 25.7–33.1) months. Lorlatinib mDOT was 5.3 (95% CI 3.6–7.6) months with a median OS from lorlatinib initiation of 14.1 (95% CI 10.3–19.2) months. The results of the brigALK2 study confirm the efficacy of brigatinib in a population of heavily pretreated *ALK*^+^ aNSCLC patients and provide new data on the activity of lorlatinib after brigatinib.

## 1. Introduction

Anaplastic lymphoma kinase (ALK) (*ALK*)-gene rearrangements ALK^+^ are oncogenic drivers found in 3–5% of patients with non-small-cell lung cancers (NSCLCs) [1]. Patients with ALK^+^ NSCLCs are generally younger and non or light smokers, with adenocarcinoma histology and a high risk of brain metastases at diagnosis and progression [1,2]. Brigatinib is a next-generation ALK inhibitor (ALKi) with enhanced brain activity and potent efficacy against many ALK-resistance mutations [3,4]. Brigatinib was initially developed for crizotinib-pretreated *ALK*^+^ NSCLCs [2,5,6].

More recently, following the results of the phase 3 trial—ALK in lung cancer trial of brigatinib in 1st line (ALTA-1L)—brigatinib was granted marketing authorization for the first-line treatment of advanced ALK^+^ NSCLCs [7,8,9,10]. ALTA-1L results indicated superior brigatinib efficacy compared to crizotinib in patients with ALK tyrosine-kinase inhibitor (TKI)-naive ALK^+^ advanced NSCLCs [7,8]: median progression-free survival (PFS) according to an independent review committee (primary endpoint) was 24.0 months with brigatinib vs. 11.1 months with crizotinib (HR 0.49 (95% confidence interval (CI) 0.35–0.68); *p* < 0.0001). At the end of the study, respective 3-year PFS determined by a blinded independent review committee was 43% vs. 19%, with a median OS not reached (NR) for either group. At present, several alternatives to crizotinib are available [10,11,12,13,14,15], rendering real-life data essential [16,17,18,19,20,21,22,23,24,25,26,27] to attempt to evaluate optimal therapeutic sequences [28].

The national, retrospective, multicenter and real-world study, BrigALK, analyzed patients with ALK^+^ advanced NSCLCs enrolled in the brigatinib French early-access program (FEAP). According to the interim BrigALK analysis, after the enrollment of 104 patients (data cut-off: 30 June 2018, median follow-up 6.7 months), the primary endpoint—invPFS from brigatinib initiation—was 6.6 months (95% CI 4.8–9.9) with median overall survival (OS) of 17.2 months (95% CI 11.0–NR) months [19].

BrigALK2 covered the entire FEAP period; an updated analysis examined brigatinib efficacy. Subsequent treatments were collected, focusing on post-brigatinib lorlatinib efficacy.

## 2. Patients and Methods

### 2.1. Study Design and Patients

The objective of this non-interventional study was to evaluate, in the real-world setting, brigatinib efficacy in ALK^+^ advanced NSCLC. Inclusion criteria were: 18 years old; *ALK*^+^ advanced NSCLCs assessed by fluorescence in situ hybridization and/or immunohistochemistry or NGS in each participating center; prior treatment with 1 ALK inhibitor(s) (ALKi(s)), including crizotinib; FEAP-provided brigatinib (1 August 2016 to 21 January 2019).

### 2.2. Data Collection

Patient data, collected retrospectively from medical files, included demographics, NSCLC characteristics, numbers and localizations of metastatic sites, previous treatments, tumor response to brigatinib, resistance mutations before starting brigatinib or after progression and post-progression treatments. All consecutive patients meeting inclusion criteria were enrolled without selection in each participating center.

### 2.3. Endpoints

The primary endpoint was invPFS from brigatinib onset, i.e., from the first brigatinib dose to first documentation of objective disease progression or death from any cause. Secondary endpoints included objective response rate (ORR), median duration of treatment (DOT), median OS and analysis of subsequent treatment(s). Analysis focused particularly on lorlatinib efficacy (median DOT and OS from lorlatinib initiation) after progression on brigatinib.

### 2.4. Statistical Analyses

Comparisons between patient characteristics were performed using the chi-square test or Fisher’s exact test for discrete variable. The Kaplan–Meier method was used to estimate PFS and OS for the entire population and defined subgroups according to the number of treatment lines. The log-rank test compared survival by treatment category. Best response to treatment was assessed according to RECIST 1.1 criteria. Statistical analyses were computed with SAS v9.4 software (SAS Institute, Cary, NC, USA).

The study was conducted in accordance with French laws and regulations in force (law 78–17 of 6 January 1978 modified by laws 94–548 of 1 July 1994, 2002-303 of 4 March 2002, 2004-801 of 6 August 2004). The GFPC has committed to the CNIL (French National Commission for Data Protection and Liberties) to respect the MR-004 reference methodology for research not involving human subjects.

## 3. Results

During FEAP period, access to brigatinib was requested for 227 patients. Among them, 183 patients managed in 66 centers were included in the BrigALK2 study. The main reasons for patient exclusion from BrigALK2 analysis are summarized in the study flowchart (Figure 1).

Patients’ characteristics before brigatinib treatment are summarized in Table 1; 37.4% had Eastern Cooperative Oncology Group performance status (ECOG PS), more than half of them (55.3%) had ≥3 metastatic sites—with the brain being the most frequent site (71.6%)—and 7.1% of patients had leptomeningeal carcinomatosis. Median time from diagnosis of metastatic NSCLC to brigatinib initiation was 30.6 months (95% CI 17–34.6). Patients received a median of 3 (range 1–6) lines and were pretreated with a median of 2 ALKis (91.8% crizotinib, 85.3% ceritinib and 9.2% alectinib) (Table 2). Brigatinib was given to 14 (7.7%) patients as second-line therapy, 44 (24%) as third line, 74 (40.4%) as fourth line and 51 (27.9%) as fifth line. Most often, therapeutic sequences comprised two (mainly, crizotinib then ceritinib, *n* = 36 (19.7%)) or three lines (mainly, chemotherapy, crizotinib then ceritinib, *n* = 49 (26.8%)); 129 (70.5%) patients had received chemotherapy before brigatinib (Appendix A). In addition, rebiopsies (tissu and ctDNA) were obtained from 51 patients (27.8%) who experienced disease progression before brigatinib treatment. Resistance-mutation genotyping was carried out for 28 patients (21 not carried out and 2 failures) and revealed a secondary mutation in 10 of them (35.7%). G1202R was the most frequently identified secondary mutations (7/10) with one case of compound mutation (F1174L/G1202R). (Type of mutation, best response to brigatinib and DOT with brigatinib are presented in Table 3.)

The standard brigatinib dose (180 mg/day) was given to 175 (95.6%) patients with a 7-day induction of 90 mg/day; 37 (20.2%) patients required dose adjustments, without definitive interruption for intolerance or patient choice, and 19 (10.4%) discontinued brigatinib because of adverse events or patient choice.

At data cut-off (1 February 2021), median follow-up was 40.4 months (95% CI 38.4–42.4) (Table 4); 21 (11.5%) patients were still on treatment. Median invPFS was 7.4 months (95% CI 5.9–9.6) and brigatinib median DOT was 7.3 months (95% CI 5.8–9.4) with 59.5% and 39% still on treatment at 6 and 12 months. For patients who received 1 (*n* = 14), 2 or 3 (*n* = 118) or >3 (*n* = 51) ALKi(s) before brigatinib, median DOTs were 13.8 (95% CI 3.8–26.4), 7.4 (95% CI 5.6–9.9) and 4.9 (95% CI 1.7–9.3) months, respectively. Final ORR and disease-control rate (DCR) for the 169 assessable patients were 44.3% and 74.2%, respectively. Median OS from brigatinib initiation in the whole population was 20.3 months (95% CI 15.6–27.4), with 78% and 63.8% alive at 6 and 12 months. As for median DOT, median OS varied according to the number of pre-brigatinib ALKis: median OS from brigatinib initiation was 33 (95% CI 9.7–NR), 20.3 (95% CI 15.7–28.7) and 18.1 (95% CI 3.3–24.5) months, respectively, for patients pretreated with 1, 2 or 3 and >3 ALKi(s). There was no significant difference in median OS between patients with and without brain metastases (20.3 months (14.7–27.4) vs. 22.6 months (12.6–37.5) respectively) (Figure 2). It should be noted that more than half of the patients had brain radiotherapy before brigatinib initiation regardless of treatment line.

At the time of the analysis, 112/183 patients (60.9%) had progressed on brigatinib. Progression involved known metastases for 110/112 (98.2%) patients (known brain lesions for 45) and new lesions for 47/112 (42%), 23 of which were cerebral. Post-brigatinib, new biopsies were obtained from 22 patients (19.6%). Genotyping, carried out for fifteen patients, identified a secondary mutation in six of them: G1202R was still the most identified secondary mutation (5/6), with another case of compound mutation (*F1174V* and *D1203N*).

Among 106 (57.9%) patients receiving at least one therapy after brigatinib, clinical data could be obtained for 92 (86.8%) of them. A total of 68 (64.1%) received lorlatinib: 51 patients (75%) received it immediately after brigatinib; 17 received it after sequential administration of another second-generation ALKi + chemotherapy (Figure 3). The interval between advanced NSCLC diagnosis and lorlatinib start was 52.8 months (95% CI 43.2–60.3). With a median follow-up of 29.9 months, the median lorlatinib DOT was 5.3 months (95% CI 3.6–7.6) and median OS from lorlatinib initiation was 14.1 months (95% CI 10.3–19.2).

Finally, the median OS from the initial NSCLC diagnosis was 75.3 months (95% CI 38.2–174.6), knowing that 86.4% of the patients had an advanced stage at diagnosis.

## 4. Discussion

With a median follow-up of 40.4 months, the results of this retrospective, multicenter, real-life study confirmed brigatinib efficacy at managing heavily pre-treated advanced ALK^+^ NSCLCs with median invPFS and OS from brigatinib initiation at 7.4 and 20.3 months, respectively. As noted previously and in another analysis [20,21,22], brigatinib effectiveness (DOT and OS) tends to decline, depending on the line at which it is used. Most patients were able to receive the standard regimen, with dose reduction for 20% of them and a 10% discontinuation rate, thereby confirming intermediate analysis findings [19].

Several recent studies examined brigatinib efficacy in heavily pretreated patients ALK^+^ advanced NSCLCs. According to a retrospective chart review of 104 brigatinib-treated patients in Italy, Norway, Spain and the UK, ORR was 39.8%, median PFS was 11.3 (95% CI 8.6–12.9) months and median OS lasted 23.3 (95% CI 16.0–NR) months [22]. Based on 604 patients (from 21 countries), including those previously given next-generation ALKis, median brigatinib DOTs for patients with prior crizotinib, alectinib, ceritinib or lorlatinib were 10.0, 8.7, 10.3 or 7.5 months, respectively [21]. Finally, a phase 2, single-arm study analyzed brigatinib efficacy and safety in 47 Japanese patients with *ALK*^+^ advanced NSCLCs that progressed from previous alectinib or other ALKis [20]. Their ORR and DCR were 34% (16/47) and 79% (37/47), respectively, with median PFS lasting 7.3 months. These summarized results differ slightly from ours and this difference is likely attributable, in part, to the limitations of retrospective chart data but also, and perhaps more importantly, to imbalances between the populations analyzed. Patients’ health status was better in the studies by Popat et al. [22] and Lin et al. [21], with respective ECOG PS 0/1 for 84% and 85.9%, and they were less intensively pretreated (median 2 lines and 1 TKI before brigatinib) in Popat et al.’s study [22] compared to medians of 3 lines and 2 TKIs herein.

Access to treatment post-brigatinib progression was also an important factor: 58% of our patients had access to post-progression therapy. Among them, nearly two-thirds received the third-generation ALKi lorlatinib with a median DOT of 5.3 months and median OS from lorlatinib initiation of 14.1 months. These results are close to those of Popat et al., who reported that 69% (53/77) of their patients who had progressed at the time of the analysis had received systemic therapy after brigatinib, 42 with an ALKi. 34 treated with lorlatinib. The median DOT for evaluable patients was 2.57 months [22].

Lorlatinib efficacy in this context was also analyzed in several studies [23,24,25,26]. Lorlatinib was accorded marketing authorization for second-line treatment after failure of a first-line second-generation TKI, regardless of the existence of a resistance mechanism. In a phase 2 trial, lorlatinib efficacy was evaluated in different populations based on treatment history before lorlatinib. Of the five cohorts, three included a second-generation TKI in their treatment sequence. Analysis of data from those three populations (comprised of 159 patients) showed an ORR of 39.6%, median PFS lasting 6.6 months and median OS 20.7 months after starting lorlatinib. These observations are quite similar to ours and are further supported by real-life data. In a multicenter retrospective analysis of lorlatinib in 37 heavily pretreated, ALK^+^ advanced NSCLC patients, median lorlatinib DOT was 4.4 months, with 43.2% ORR and median OS from lorlatinib onset lasting 10.2 months [24]. Another analysis of 22 patients in the same setting found 35.7% ORR and 64.3% DCR, with median PFS at 6.2 months. PFS was longer for patients who benefited from prior ALKi(s) than those who did not (6.5 vs. 3.5 months, respectively) [24]. In a real-world analysis of 76 *ALK*^+^ NSCLC patients enrolled in early or expanded access programs for lorlatinib in Asia and the United States, respective ORR and median PFS for those treated with <2 previous TKIs, 2 previous TKIs and 3 previous TKIs, were 42% and NR months, 35% and 11.2 months and 18% and 6.5 months [27].

There is a growing body of literature on the search for mechanisms of resistance to ALKi but this practice is not mandatory for progression management [28,29,30]. In our retrospective series, rebiopsy was performed in approximately 25% and 20% of pre- and post-brigatinib patients. Though these rates may seem low, they represent an accurate reflection of a period of management of ALK+ aNSCLC that corresponds to the brigatinib EAP between August 2016 and January 2019. When genotyping was performed, it led to the identification of an ALK secondary mutation in 37.5% and 40%, pre- and post- brigatinib, respectively, but we could not know whether the identification of the mutations was considered for the therapeutic decision. These rates of resistance mutations are demonstrated in a series of patients pre-treated with a median of two pre-brigatinib TKIs, mainly the crizotinib-ceritinib sequence. They are quite similar to those reported by Gainor et al. [31]: after two TKIs, including a second-generation TKI, secondary mutations were detected in approximately 50% of cases. Moreover, among mutations detected when patients experienced progression, G1202R was the most common. This is also consistent with previously known data that have shown the high frequency of the G1202R mutation after second-generation TKIs [32].

Our study has several limitations. First of all, it should be noted that treatment sequences are currently different from those analyzed in the brigALK study, as most patients are treated with a second-generation TKI in the first line (alectinib or brigatinib). Other limitations are those inherent in this type of retrospective study without data monitoring. Treatment efficacy was assessed by the investigators, without any independent review committee. Investigator assessment bias cannot be excluded. In this real-life study, it was not always possible to obtain complete information from patients’ medical records. That was the case, for example, for patients with a dose reduction or those who discontinued brigatinib before disease progression. Another limitation to point out is the small number of events on lorlatinib that could explain the CI 95% of DOT and OS that are a little too wide. One of the strengths of the study is the absence of stringent criteria for study inclusion, meaning the population is representative of real-life, heavily treated patients with *ALK*^+^ advanced NSCLCs.

## 5. Conclusions

This analysis of FEAP data confirmed brigatinib effectiveness in a population of heavily pretreated ALK^+^ advanced NSCLCs and provided informative real-life observations about lorlatinib efficacy post-brigatinib.

## Figures and Tables

**Figure 1 cancers-14-01751-f001:**
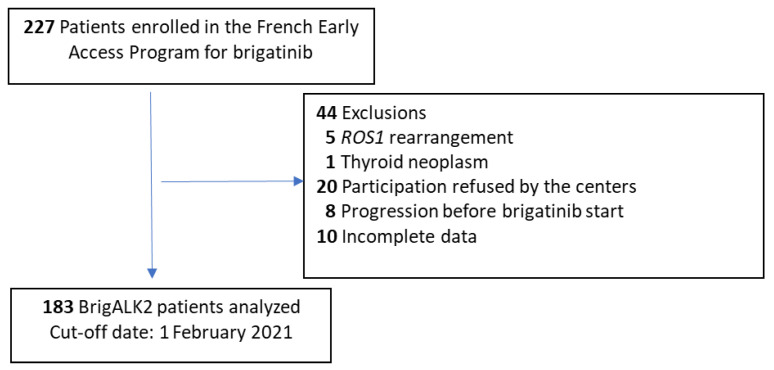
Study flow-chart.

**Figure 2 cancers-14-01751-f002:**
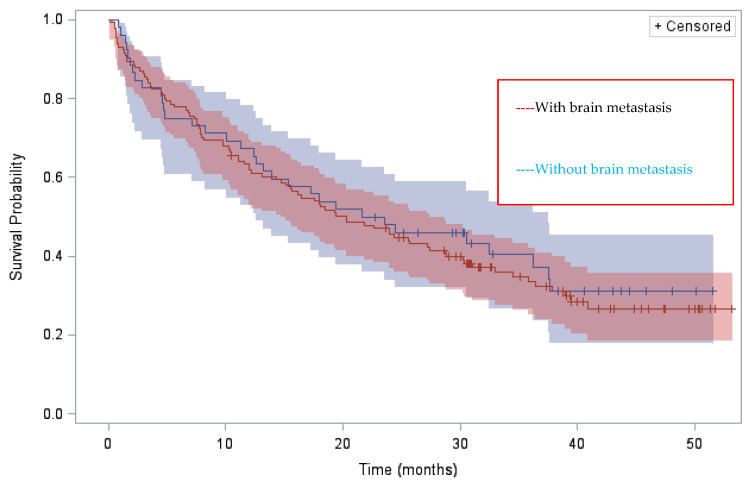
Median overall survival from brigatinib initiation of patients with and without brain metastases.

**Figure 3 cancers-14-01751-f003:**
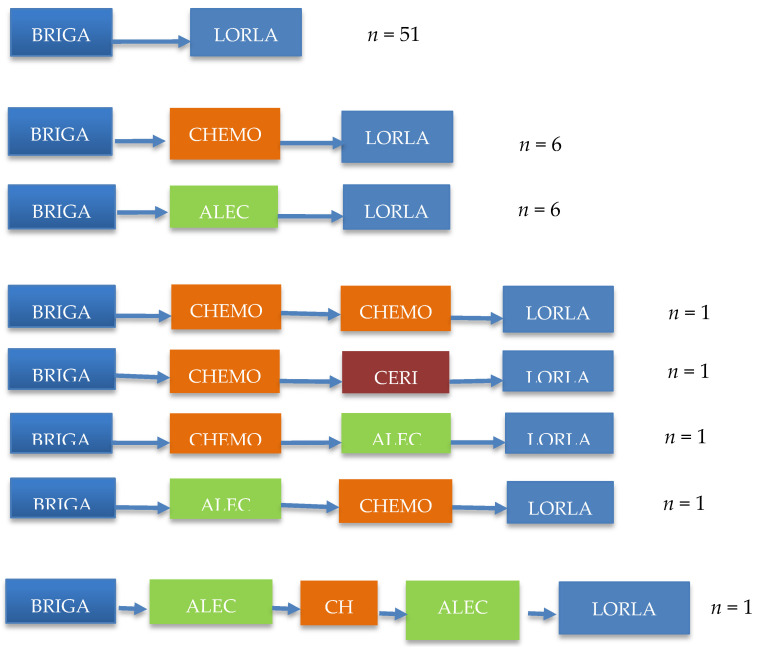
Therapeutic sequences after brigatinib.

**Table 1 cancers-14-01751-t001:** Characteristics of the 183 patients with ALK^+^ advanced NSCLCs before brigatinib.

Characteristic	Value
Age, median ± IQR years	60 ± 12.7
Sex	
Male	75 (41)
Female	108 (59)
Smoking status	
Current smoker	30 (16.4)
Never/former smoker	153 (83.6)
Histology	
Adenocarcinoma	178 (97.3)
Others	5 (2.7)
ECOG performance score (*n* = 147)	
0	38 (25.9)
1	54 (36.7)
≥2	55 (37.4)
General symptoms	114 (62.3)
Neurological and meningeal symptoms	47 (25.7)
Weight loss >5% (*n* = 108)	44 (40.7)
Number of metastatic sites (*n* = 180)	
1	36 (20)
2	44 (24.4)
>2	100 (55.6)
Metastatic site locations (*n* = 180)	
Central nervous system	131 (72.8)
Lung	63 (35)
Pleura	34 (18.9)
Liver	48 (26.7)
Bone	82 (45.6)
Leptomeningeal carcinomatosis	13 (7.2)
Adrenals	22 (12.2)
Pericardium	7 (3.9)
Others	66 (36.7)
Brain radiotherapy	
1st line	92 (50.2)
2nd line	21 (11.5)
3rd line	13 (7.1)
4th line	9 (4.9)
Before brigatinib	13 (7.1)

**Table 2 cancers-14-01751-t002:** Systemic treatment modalities given to the 183 patients with ALK+ advanced NSCLCs before brigatinib.

Treatment History	*n* (%)
Previous lines received, *n*	
1	15 (8.2)
2	47 (25.7)
3	74 (40.4)
4	26 (14.2)
>5	22 (12)
First-line modality (*n* = 183)	
Chemotherapy	100 (54.6)
Crizotinib	78 (42.6)
Alectinib	5 (2.7)
Second-line modality (*n* = 169)	169 (92.3)
Chemotherapy	25 (13.7)
Crizotinib,	83 (45.3)
Ceritinib	55 (30.1)
Alectinib	5 (2.7)
Immunotherapy	1 (0.5)
Third-line modality (*n* = 122)	122 (72.6)
Chemotherapy	25 (20.5)
Crizotinib	10 (8.2)
Ceritinib	76 (62.3)
Alectinib	4 (3.3)
Agent received just before brigatinib	183 (100)
Chemotherapy	27 (14.7)
Crizotinib	27 (14.7)
Ceritinib	114 (61.6)
Alectinib	9 (4.9)
Lorlatinib	6 (3.3)
Immunotherapy	1 (1.1)
Number progressing before brigatinib	179
1	34 (18.6)
2	49 (26.8)
3	60 (32.8)
4	21 (11.5)
>5	15 (8.2)

**Table 3 cancers-14-01751-t003:** Secondary mutations identified before brigatinib treatment (*n*= 10): best response and DOT with brigatinib.

Patient	Type of Mutation	TKI before Brigatinib	Best Response to Brigatinib	DOT with Brigatinib (Months)
1	G1202R	Crizotinib	PR	5.2
2	G1202R	Ceritinib	PD	1.5
3	G1202R	Ceritinib	SD	5.5
4	G1202R	Alectinib	PR	9.2
5	G1202R	Crizotinib	PR	7.5
6	G1202R	Ceritinib	SD	17.5
7	F1174L/G12020R	Missing data	PD	1
8	L861Q	Alectinib	PR	9.2
9	G1269A	Chemotherapy	PD	2.8
10	C1156Y	Crizotinib	PR	52.2

PR, partial response; PD, progression disease; SD, stable disease; DOT, duration of treatment.

**Table 4 cancers-14-01751-t004:** Outcome parameters for the 183 brigatinib-treated patients.

Outcome Parameter	Months (95% CI)
Brigatinib-treated patients (*n* = 183)	
Median follow-up	40.4 (38.4–42.4)
Median investigator assessed PFS	7.4 (5.9–9.6)
Median DOT, brigatinib	7.3 (5.8–9.4)
1 ALKi (*n* = 22)	13.8 (3.8–26.4)
2 ALKis (*n* = 146)	7.4 (5.6–9.9)
3 ALKis (*n* = 15)	4.9 (1.7–9.3)
Median OS from brigatinib start	20.3 (15.6–27.6)
1 ALKi (*n* = 22)	33 (9.7—not reached)
2 ALKis (*n* = 146)	20.3 (15.7–28.7)
3 ALKis (*n* = 15)	18.1 (3.3–24.5)
Lorlatinib post-brigatinib patients (*n* = 68)	
Median DOT, lorlatinib	5.3 (3.6–7.6)
Median OS, lorlatinib	14.1 (10.3–19.2)

PFS, progression free survival; DOT, duration of treatment; ALKi, ALK inhibitor; OS, overall survival.

## Data Availability

The data presented in this study are available on request from the corresponding author.

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
