# Peer review of "Brigatinib for Pretreated, ALK-Positive, Advanced Non-Small-Cell Lung Cancers: Long-Term Follow-Up and Focus on Post-Brigatinib Lorlatinib Efficacy in the Multicenter, Real-World BrigALK2 Study"

_cancers, 2022, doi:10.3390/cancers14071751_

Round 1
Reviewer 1 Report
Please check:
line 131: 37.1% had PS 2, but in the table 1 is 37.4 (ps>2), please verify.
line 133, maybe is >3 metastatic sites (and not 3)or if it is 3, in table 1 we have >2, please clarify
Reviewer 2 Report
In this paper the authors assess the efficacy of two next-generation ALK inhibitors, Brigatinib and Lorlatinib in pretreated advanced NSCLC. The patient cohort is from the French Early Access Program, with approximately 40% patients having an ECOG >1. The article is soundly written and the data are of great interest to the community. Apart from minor points that should be addressed before publication I have nothing to add.
Minor
1. line 189: The median OS since diagnosis of NSCLC was 75.3 months. Could the authors please provide data on the initial stages?
2. line 173: No difference in median OS between patients with and without brain metastases could be seen. Could the authors please refer to the role of brain radiotherapy for brain metastases, which was the 1st line therapy for 50% of the patient cohort. Would the authors recommend omitting the routine use of brain RT before brigatinib and keep it as a salvage option?
3. Table 1 Number of metastatic sites 100 (55/.6) - what does this mean?
Reviewer 3 Report
This report is the evaluation of efficacy of brigatinib for ALK+NSCLC patients who received at least 1 ALKi therapy with using real-world data. I completely agree with that the data is so valuable to be published. To be better for readers, I provide some suggestion and comments.as below.
- In the last sentence of the Simple summary in page1, the term “(journal name)” revised to “Cancers”.
- In the abstract, the first sentence,”Brigatinib is a next-generation ALK inhibitor (ALKi) approved in ALK inhibitor naïve and post-crizotinib ALK+ advanced NSCLCs (aNSCLCs).” should be included the location for the approval, such as “in French”. There it a country which approve brigatinib as second-line therapy regardless of prior ALKi, not only crizotinib but alectinib or ceritinib. Or, author can use the term that express the efficacy such as “establish evidence” or “show efficacy”, instead of “approve”
- In Table 3 of page 6, if the type or mutation in the patient 6 and 7 is wrong, do revise G12020R into G1202R.
- In line of 246, the authors mentioned that the search for a mechanism of resistance is recommended. To my best knowledge, whether AKLi-selection based on the secondary mutation as mechanism of resistance is effective or not has not yet been known. I feel that it is slightly overstatement, and I recommend that the author revise the term “recommend”. If guideline recommend the rebiopsy after ALKi failure, the author should describe the reference.
- The rate of re-biopsy, approximately 25% and 20% of pre- and post-brigatinib patients, seems to be higher than that in clinical practice. We has not yet proven the treatment strategy based on secondary mutation after ALKi failure unlike EGFR mutation, therefore we do not usually perform re-biopsy for ALK-positive patients who experience PD on ALKi therapy. Was the re-biopsy prospectively planned to be performed in the brigatinib French Early-Access Program? The aurhors may as well describe the reason of higher rate of re-biopy for the readers who feel the discrepancy against clinical practice in their counties.
- How about the efficacy of chemotherapy in this study, especially after brigatinib failure in chemotherapy-naïve patients? Through these data, what we medical physicians really want to know is when is best timing of administration of chemotherapy. The authors mentioned that they focus the efficacy of lorlatinib after brigatinib, and I also agree with that the data could provide us the hint about the treatment strategy based on the information of mechanism of resistance, in addition it can be also rationale for usage of lorlatinib after brigatinib failure. However, DOT of lorlatinib after brigatinib was 5.3 months in this study, and I guess that PFS of chemotherapy could be more effective for chemotherapy-naïve patients. DOT has trend to be prolonged especially in late-line therapy due to continuous usage beyond PD in the analysis using real-world data, which can be limitation of this study. The authors focus on the efficacy of lorlatinib after brigatinib, so I recommend that the authors discuss whether lorlatinib after brigatinb therapy is really efficacy compared to other treatment option.
- When seeing the 95%CI of DOT or OS of the lorlatinib-analysis, they seems to be too wide even with consideration of small sample size. I guess that the 95%CI may be wide due to the small number of events. If so, I suggest that the authors add the small number events as one of the limitations.
- The authors used Kaplan-Meier method for estimating PFS and OS. I recommend that Kaplan-Meier curves as figure should be described in the manuscript.
Reviewer 4 Report
Thank your for your valuable work. However, there are some flaws to be addressed.
-Each table and figure should have a legend in order to explain the acronyms
Round 2
Reviewer 3 Report
I checked the authors’ reply and revised manuscript, and I agree with the revision as the response to my review comments.